# Distance estimation from monocular cues in an ethological visuomotor task

**Philip RL Parker[1]\*, Elliott TT Abe[1], Natalie T Beatie[1], Emmalyn SP Leonard[1], Dylan M Martins[1], Shelby L Sharp[1], David G Wyrick[1], Luca Mazzucato[1,2], Cristopher M Niell[1,3]\***

[1]Institute of Neuroscience, University of Oregon, Eugene, United States; [2]Department of Mathematics, University of Oregon, Eugene, United States; [3]Department of Biology, University of Oregon, Eugene, United States

**\*For correspondence:**
prlparker@gmail.com (PRLP);
prlparker@gmail.com (PRLP);
cniell@uoregon.edu (CMN)

**Abstract** In natural contexts, sensory processing and motor output are closely coupled, which is reflected in the fact that many brain areas contain both sensory and movement signals. However, standard reductionist paradigms decouple sensory decisions from their natural motor consequences, and head-fixation prevents the natural sensory consequences of self-motion. In particular, movement through the environment provides a number of depth cues beyond stereo vision that are poorly understood. To study the integration of visual processing and motor output in a naturalistic task, we investigated distance estimation in freely moving mice. We found that mice use vision to accurately jump across a variable gap, thus directly coupling a visual computation to its corresponding ethological motor output. Monocular eyelid suture did not affect gap jumping success, thus mice can use cues that do not depend on binocular disparity and stereo vision. Under monocular conditions, mice altered their head positioning and performed more vertical head movements, consistent with a shift from using stereopsis to other monocular cues, such as motion or position parallax. Finally, optogenetic suppression of primary visual cortex impaired task performance under both binocular and monocular conditions when optical fiber placement was localized to binocular or monocular zone V1, respectively. Together, these results show that mice can use monocular cues, relying on visual cortex, to accurately judge distance. Furthermore, this behavioral paradigm provides a foundation for studying how neural circuits convert sensory information into ethological motor output.

## Editor's evaluation

This is an important article with compelling experimental methods, including ethologically relevant behavior, sophisticated physiological methods including optogenetic suppression of primary visual cortical activity, careful behavioral experiments, and clear, convincing, quantitative analysis of the resulting data. The article enhances our understanding of the role of active visual estimation of distance under multiple factors of visual degradation (binocular/monocular, and V1 suppression), demonstrating how robust task performance can emerge from compensatory active sensing strategies.

## Introduction

Vision is an active process – we continuously move our eyes, head, and body to gain information about the world around us. One core function of active vision is to determine the distance between the observer and objects in its environment. This ability is so critical that many species have evolved to use multiple distinct cues to estimate depth, including retinal image size, motion and position parallax, and binocular disparity (*Kral, 2003*; *Shinkman, 1962*). In particular, depth perception

through stereo vision has been heavily studied, but other cues that provide important complements are less well understood. Furthermore, some of these monocular cues, such as motion parallax and loom, are closely integrated with movement. How does the brain make use of these diverse cues to guide different behaviors? For instance, is distance explicitly computed and represented in neural activity for some behaviors and implicitly encoded for others? Furthermore, how is this sensory representation converted into the appropriate motor output? Neurophysiological studies are often performed on head-fixed subjects, limiting the range of depth cues and behaviors that can be studied. Addressing these questions requires behaviors where experimental subjects amenable to neural circuit interrogation can engage in distance estimation behaviors unrestrained (*Leopold and Park, 2020*; *Parker et al., 2020*).

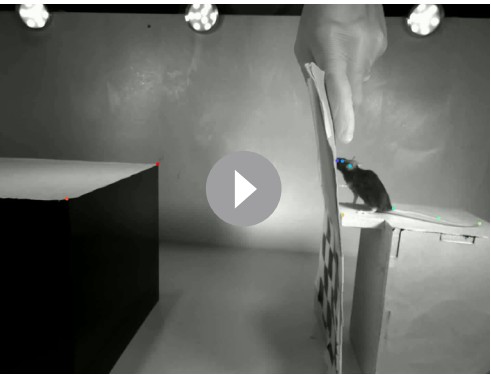

**Video 1.** Mouse performing the task under binocular conditions with DeepLabCut labels overlaid.
https://elifesciences.org/articles/74708/figures#video1

The mouse is an important model for vision, yet relatively few behavioral paradigms exist for studying natural, active vision in the mouse (*Boone et al., 2021*; *Hoy et al., 2016*; *Yilmaz and Meister, 2013*). Previous work in other rodent models, including rats and gerbils, showed that animals will accurately jump to distant platforms for a reward, and that changing experimental conditions can bias animals toward the use of certain depth cues, including monocular ones (*Carey et al., 1990*; *Ellard et al., 1984*; *Goodale et al., 1990*). Here, we report that mice are capable of using vision to estimate the distance across a variable gap and execute an accurate ballistic jump. Using this behavior, we show that mice can use monocular vision to judge distance, and suppressing the activity of primary visual cortex (V1) disrupts task performance. Furthermore, this paradigm provides a foundation for studying various visual computations related to depth, and the corresponding motor output, in a species amenable to measurement and manipulation of neural activity in genetically defined populations.

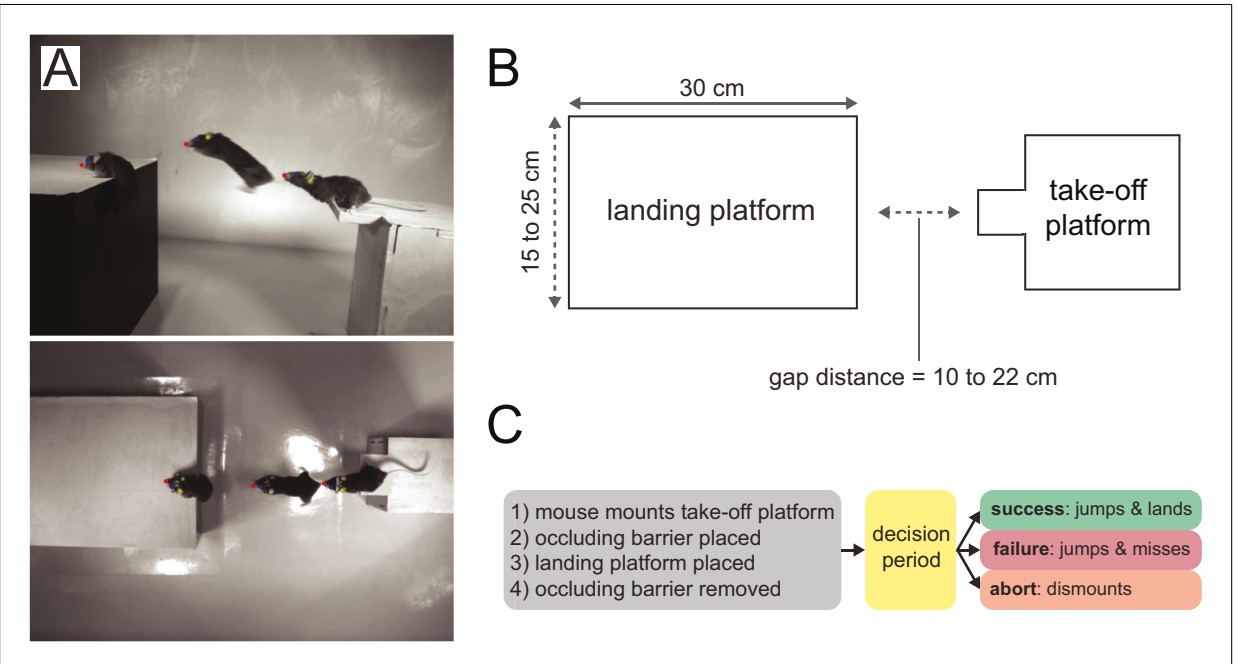

**Figure 1.** Mouse distance estimation (jumping) task. (**A**) Example side and top-down video frames (three overlaid) from a single trial. (**B**) A random combination of landing platform width (three sizes) and gap distance (seven distances) is chosen for each trial. (**C**) Trial logic.

## Results

### Mouse distance estimation (jumping) task

In order to establish a framework for studying distance estimation in the mouse, we adapted a gerbil/rat jumping task (*Ellard et al., 1984*; *Legg and Lambert, 1990*; *Richardson, 1909*), where animals were rewarded for successfully jumping across a variable gap (*Figure 1A*). Mice were free to roam around the arena, then initiated trials by mounting a take-off platform. An occluding barrier was introduced to block the mouse's view while the experimenter randomly placed one of three landing platforms at one of seven distances from the take-off platform (*Figure 1*). We used landing platforms of variable size to minimize the use of retinal image size cues, which may not require visual cortex for accurate distance estimation after learning (*Carey et al., 1990*). The trial began as soon as the occluding barrier was removed, and the decision period comprised the time between barrier removal and the last video frame before the mouse executed one of three outcomes (*Figure 1C*). On 'success' trials, the mouse jumped and landed on the landing platform, and received a reward (see *Video 1*). On 'failure' trials, the mouse jumped and missed the landing platform, landing on the arena floor, and received no reward. On 'abort' trials, mice dismounted the take-off platform onto the arena floor and received a mild air puff. Training, which usually took one to two weeks, was complete when mice successfully jumped to each of the three landing platforms at the maximum distance (22 cm). To quantify behavior, markerless pose estimation was performed on side- and top-view video with DeepLabCut (DLC; *Mathis et al., 2018*).

### Mice accurately estimate distance under binocular and monocular conditions

Mice successfully jumped to all three sizes of platforms at all gap distances (*Figure 2A*, example, blue lines in B top; N = 3580 trials in eight mice), with only a minor effect of gap distance on success rate (ANOVA, $F$ = 2.316, p=0.048). On success trials, the distance jumped increased as a function of gap distance (*Figure 2B*, blue line in bottom panel; ANOVA, distance $F$ = 12.845, p=1.17e-8), showing that mice accurately jumped rather than adapting an alternative strategy (e.g., picking one of two jump forces across the five distances). The gap was too large for mice to reach across with their whiskers, preventing the use of somatosensation to judge the distance. Furthermore, mice did not perform any jumps in the dark (n = 4 mice, four sessions), suggesting that they relied on vision.

A number of depth cues are available in natural contexts. To test the need for stereopsis, we performed monocular eyelid suture (N = 1613 trials in eight mice), after which mice performed equally well at the task with no significant difference in the fraction of success, failure, and abort trials (*Figure 2B*, magenta lines, and *Figure 2—figure supplement 1*; ANOVA binocular vs. monocular, failure $F$ = 0.002, p=0.965, success $F$ = 0.101, p=0.752, abort $F$ = 0.275, p=0.601). There was no effect of gap distance on success (ANOVA, $F$ = 1.593, p=0.169), and mice similarly increased distance jumped as a function of gap distance under monocular conditions (ANOVA, $F$ = 5.623, p=1.68e-4). These data suggest that binocular vision is not required for accurate distance estimation under these conditions and demonstrate that mice can use monocular cues to accurately judge distance. We also tested for a role of retinal image size by analyzing performance across the three different landing platforms (*Figure 2C*). Mice performed similarly across all three sizes, suggesting that they did not rely primarily on retinal image size, although distance jumped was influenced by platform size under monocular conditions (ANOVA; success, binocular $F$ = 2.345, p=0.099, monocular $F$ = 0.774, p=0.463; distance jumped, binocular $F$ = 4.436, p=0.013, monocular $F$ = 3.261, p=0.041). Finally, to determine whether accuracy or precision were altered after monocular occlusion, we calculated the change in the mean landing position between the two conditions, and the standard deviation of the landing position in each condition (*Figure 2D*). The standard deviation of landing position was similar across conditions and was greater than the difference in the mean landing position, suggesting that mice were equally accurate and precise after monocular occlusion.

### Mice perform more head movements under monocular conditions

To quantify the fine-scale structure of behavior leading up to the jump, we analyzed both the movement and position of the mouse during the decision period (*Figure 3A*) as differences between binocular and monocular conditions could indicate a change in the use of visual cues. To analyze

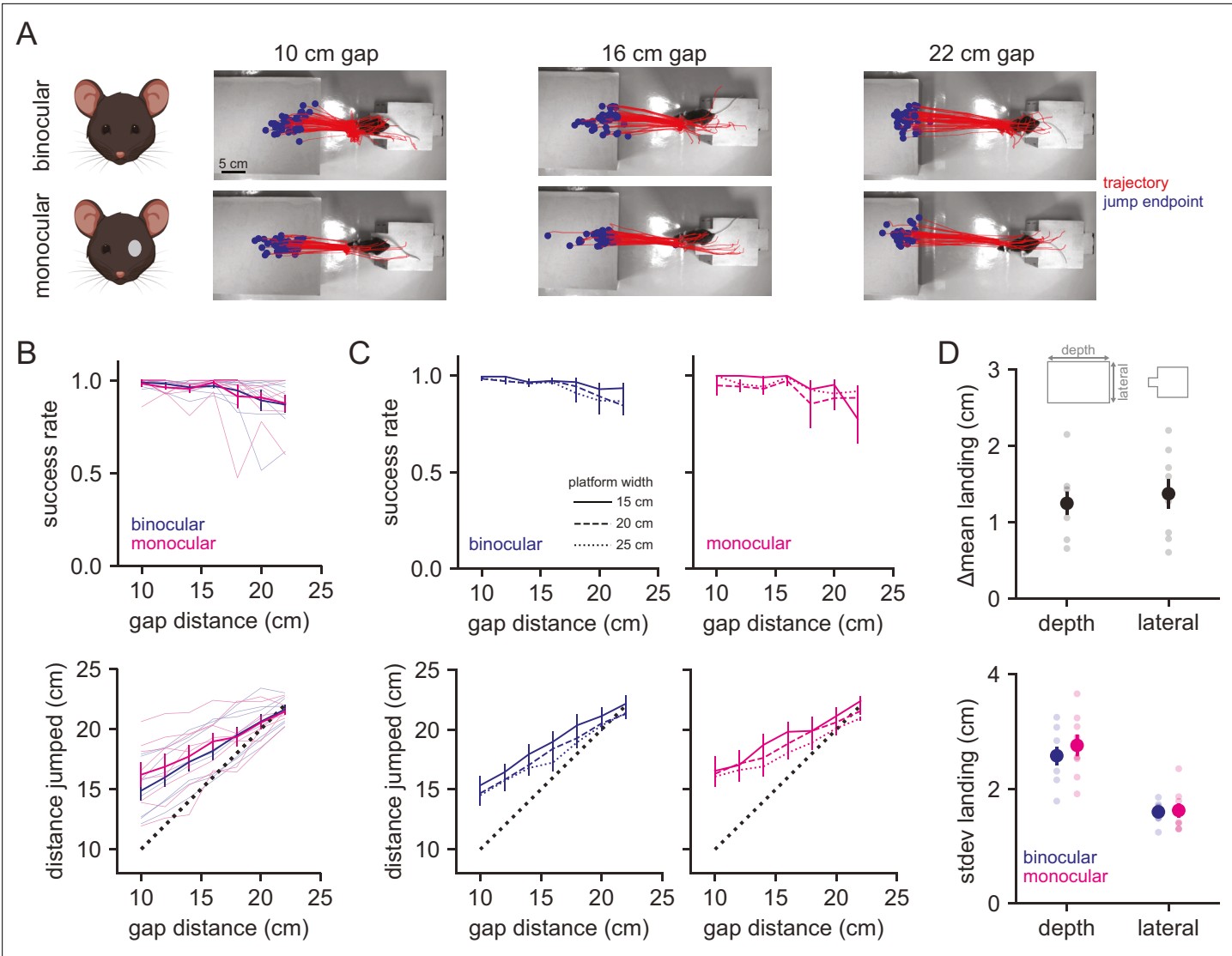

**Figure 2.** Mice accurately judge distance under binocular and monocular conditions. (**A**) Example jump trajectories from a single mouse (red line is trajectory of left ear tracked by DeepLabCut, blue dot is end point of jump) at three distances for binocular (top row) and monocular (bottom row) trials. (**B**) Performance (top) and accuracy (bottom) in binocular (blue, n = 8 mice) and monocular (magenta, n = 8 mice) conditions averaged across landing platform widths. Thin lines are individual animal data. (**C**) Performance (top) and distance jumped (bottom) for bi/monocular conditions by landing platform width (indicated by line style). (**D**) Change in the mean landing position (top) and standard deviation of landing position (bottom) for binocular vs. monocular conditions. Smaller points are individual animal data.

The online version of this article includes the following figure supplement(s) for figure 2:

**Figure supplement 1.** Binocular vs. monocular task performance.

movements, we identified zero crossings in the velocity of eye position from the side-view camera data, then took a 500 ms window around these time points, discarding any with vertical amplitudes less than 1 cm (*Figure 3B*). We then performed principal component analysis on the concatenated x/y traces, and k-means clustering on the reduced data (k = 10, see 'Materials and methods' for details). The resulting movement clusters (*Figure 3C*; ordered by total variance, high to low) showed a diversity of trajectories that together capture most of the head movements that the mice made leading up to the jump (example clusters in *Figure 3A and B*; see *Video 2*). The average trajectories of these movement clusters were highly similar between the binocular and monocular conditions (*Figure 3C*). The frequency of movements per trial was significantly increased under monocular conditions across clusters (*Figure 3D*; ANOVA; binocular vs. monocular, $F = 16.633$, p=7.58e-5), though no individual cluster showed a significant increase after accounting for repeated measures (p>0.005). Both the

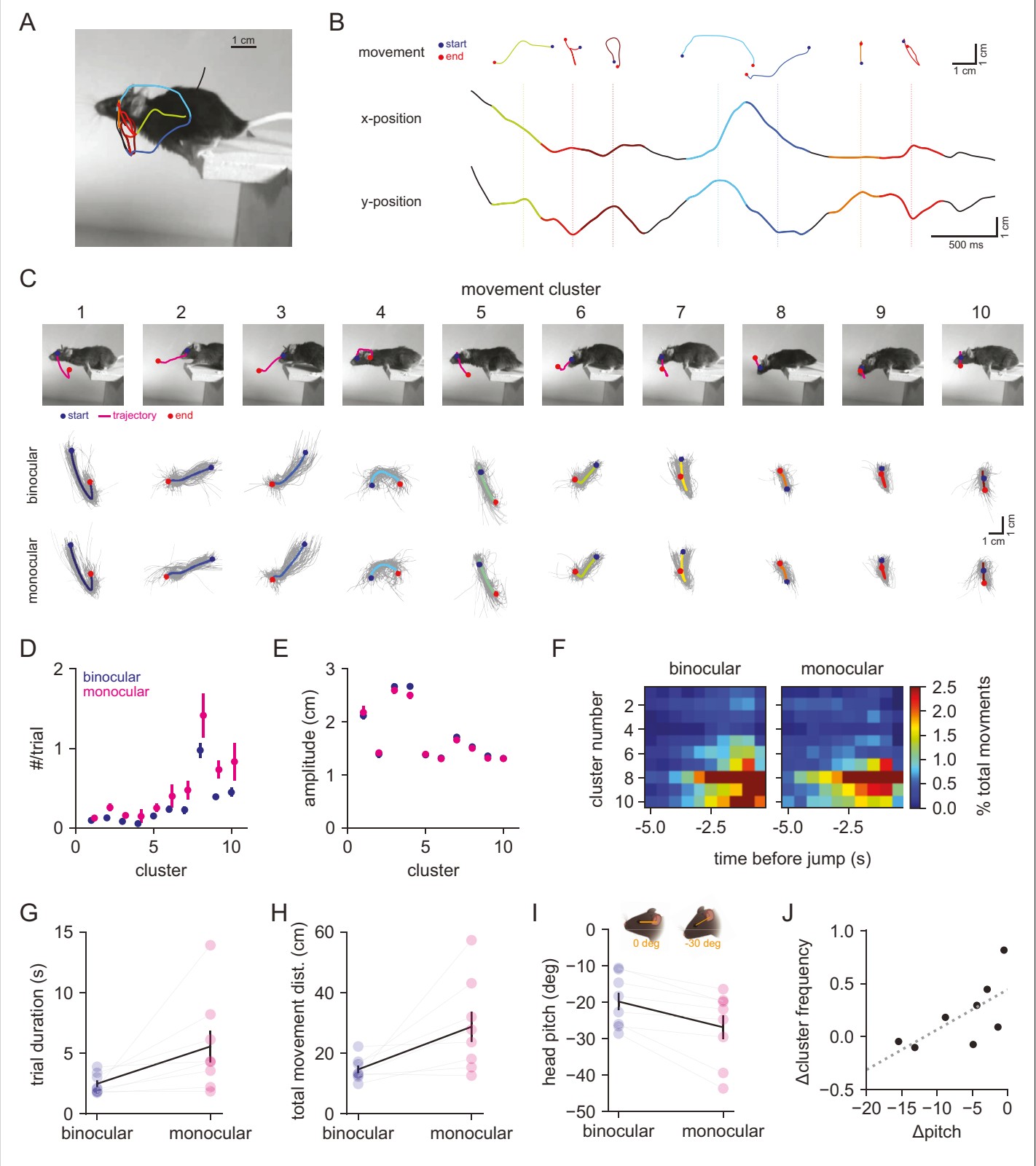

**Figure 3.** Mice perform more head movements during the decision period under monocular conditions. (**A**) Example decision period trajectory of the left eye position overlaid with movements identified through velocity zero crossings. Color corresponds to movement cluster identity in (**B ,C**). Image is the last time point in the decision period. (**B**) Horizontal (x) and vertical (y) positions of eye across time from the trace in (**A**). Individual movements are plotted above as x/y traces, with dotted lines corresponding to the middle time point, and blue and red points indicating the start and end, respectively.

*Figure 3 continued on next page*

*Figure 3 continued*

Colors correspond to clusters in (**C**). (**C**) Top: example individual movements from 10 k-means clusters; magenta is the trajectory, blue and red are start and end points, respectively. Bottom: individual movement clusters for binocular (top row) and monocular (bottom row) conditions, with means plotted over 100 individual examples in gray. (**D**) Mean number of movements per trial for each cluster in binocular (blue) vs. monocular (magenta) conditions. (**E**) Mean amplitude of movement clusters for binocular (blue) and monocular (magenta) conditions. (**F**) Normalized movement frequency as a function of time before the jump for all clusters. (**G**) Mean trial duration (decision period only) for the two conditions. (**H**) Mean of the total distance traveled by the eye during the decision period for the two conditions. (**I**) Mean head pitch, measured as the angle between the eye and ear, across the decision period for the two conditions. (**J**) Relationship between the change in head pitch and change in cluster frequency between the binocular and monocular conditions. Dotted line is fit from linear regression.

The online version of this article includes the following figure supplement(s) for figure 3:

**Figure supplement 1.** Autoregressive hidden Markov(ARHMM) modeling of decision period behavior.

amplitude (*Figure 3E*; ANOVA; binocular vs. monocular *F* = 1.349, p=0.247) and relative timing (*Figure 3F*) of movement clusters were unchanged. We confirmed that movement frequency per trial was increased in an additional dataset by performing autoregressive hidden Markov modeling on the nose, eye, and ear positions, and found that binocular and monocular conditions could be differentiated by a simple decoder using the transition probabilities between movement states (*Figure 3—figure supplement 1*).

To determine whether the increased frequency of head movements reflected an increase in sampling rate, or an increase in temporal integration, we compared the decision period duration (*Figure 3G*) and total distance moved (*Figure 3H*). Monocular animals spent more time making the decision to jump (Wilcoxon signed-rank test; binocular 2.48 ± 0.27 s vs. monocular 5.56 ± 1.31 s, p=0.008) and moved a greater distance overall than binocular animals (Wilcoxon signed-rank test; binocular 14.65 ± 1.27 cm vs. monocular 28.73 ± 5.01 cm, p=0.016). These results suggest that mice use a temporal integration strategy when binocular cues are unavailable.

Finally, to determine whether animals changed their head position under monocular conditions, we calculated head angle from the side (pitch) and top (yaw) camera data. On average, mice decreased the pitch of their head (downward tilt) under monocular conditions (*Figure 3I*; Wilcoxon signed-rank test; binocular –19.76 ± 2.39° vs. monocular –26.84 ± 3.28°, p=0.008) without changing the range (standard deviation) of pitch values (data not shown; Wilcoxon signed-rank test; binocular 15.25 ± 0.81° vs. monocular 16.79 ± 0.72°, p=0.109). Interestingly, the mean change in pitch was inversely correlated with overall change in cluster frequency (*Figure 3J*; Spearman correlation, p=0.047), suggesting that mice either increase the frequency of vertical head movements or change the positioning of their head. Yaw was not significantly changed (data not shown; Wilcoxon signed-rank test; binocular 17.15 ± 2.44° vs. monocular 17.45 ± 2.93°, p=0.641) while the range of yaw values was significantly increased (data not shown; Wilcoxon signed-rank test; binocular 8.33 ± 1.45° vs. monocular 16.24 ± 4.16°, p=0.039), suggesting an overall increase in the range of side-to-side head movements. These changes were not associated with changes in movement cluster frequency (data not shown; Spearman correlation, p=0.320).

Together, these results show that movement and position of the head during distance estimation differ when binocular vision is no longer available, consistent with a switch from the use of binocular cues to other cues such as motion and/or position parallax that require temporal integration.

## Eye movements compensate for head movements to stabilize gaze

Previous work shows that the majority of eye movements in rodents, including mice, are compensatory for head movements, and that saccades occur primarily as a consequence of large-amplitude head movements (*Meyer et al., 2020*; *Michaiel et al., 2020*; *Wallace et al.,*

**Video 2.** Same trial as *Video 1*, but with a 500 ms history of eye position labeled, along with cluster identities of movements and trial events.
https://elifesciences.org/articles/74708/figures#video2

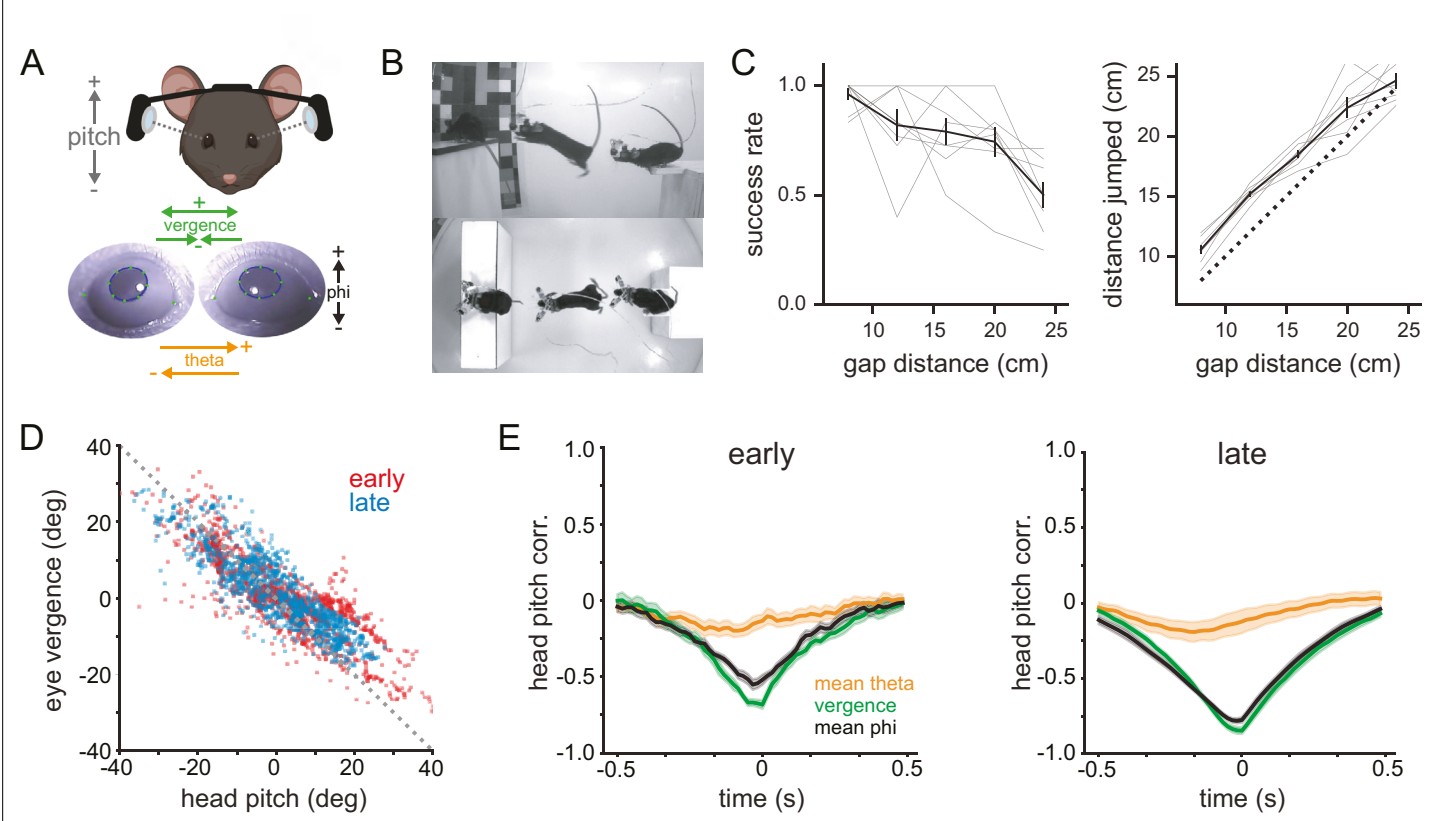

**Figure 4.** Eye movements compensate for head movements to stabilize gaze. (**A**) Schematic of experimental setup for measuring head and eye movements; bilateral eye tracking with miniature head-mounted cameras (top) and ellipse fitting of DLC-tracked pupil points (bottom). (**B**) Side and top-view images of a mouse performing the task with the eye tracking system (three frames overlaid). (**C**) Performance (left) and distance jumped (right) for eye-tracking experiments. Gray lines are individual animal data. (**D**) Horizontal angle between the two eyes (eye theta divergence) as a function of head pitch during the decision period. 'Early' is from the start of the trial to 2 s before the jump, and 'late' is the 2 s preceding the jump. (**E**) Mean eye theta, eye theta vergence, and eye phi cross-correlations with head pitch angle for early (left) and late (right) portions of the decision period; n = 8 mice for all plots.

2013). Some species make horizontal vergence eye movements to increase binocular overlap during behaviors such as prey capture (*Bianco et al., 2011*). To determine how mice target their gaze during binocular distance estimation, we performed bilateral eye tracking using miniature head-mounted cameras (*Meyer et al., 2018*; *Michaiel et al., 2020*; *Sattler and Wehr, 2020*), then used DLC to track the pupil in order to quantify horizontal and vertical eye movements (*Figure 4A*; see *Video 3*). Importantly, mice continued to accurately perform the task despite the head-mounted hardware and tether (~3 g weight; *Figure 4B and C*). It should be noted that these experiments were performed in a different set of animals with narrower landing platforms than the other experiments in this study, so performance is worse relative to the data in *Figure 2*; however, subjects showed no difference in their performance relative to their performance under control conditions (ANOVA; control vs. eye cameras $F = 0.373$, p=0.543). Head pitch (vertical head angle) was anticorrelated with both eye vergence (horizontal angle of the two eyes) and eye phi (vertical eye movements) both during the early portion of the decision period when mice were approaching the jump (start of trial to 2 s before jump; pitch vs. vergence $R^2 = 0.51$, phi $R^2 = 0.28$) and in the late portion of the decision

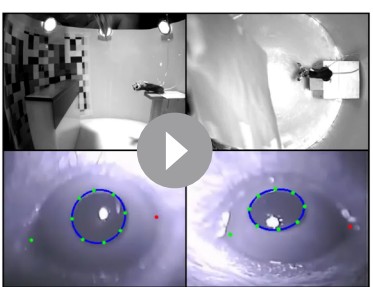

**Video 3.** Mouse performing the task with miniature head-mounted cameras tracking both eyes.
https://elifesciences.org/articles/74708/figures#video3

period immediately prior to the jump (2 s prior to jump; pitch vs. vergence $R^2$ = 0.70, phi $R^2$ = 0.54; *Figure 4D and E*). Thus, upward head movements caused the eyes to move down and toward the nose, while downward head movements caused upward and outward eye movements, consistent with vestibulo-ocular reflex-mediated gaze maintenance throughout the decision period. Additionally, while there was a slight change in vergence between the early and late periods of the trial (vergence early 1.77 ± 0.21°, late –1.66 ± –0.29°; p=1.57e-4), this was explained by a similar difference in head pitch between these two periods (pitch early –45.45 ± 1.24°, late –38.47 ± 1.43°; p=4.97e-5), demonstrating that mice do not move their eyes to increase binocular overlap preceding the jump.

## V1 optogenetic suppression disrupts distance estimation task performance

Finally, we tested whether distance estimation behavior requires visual cortex. We first asked whether suppressing the activity of binocular zone V1, which corresponds retinotopically to the central visual field in front of the mouse, affected task performance. Bilateral optic fibers were implanted at the surface of the cortex above binocular V1 in either control mice (PV-Cre) or mice expressing channelrhodopsin-2 (ChR2) in parvalbumin-expressing inhibitory interneurons (PV-Cre:Ai32, referred to as PV-ChR2 here; *Hippenmeyer et al., 2005*; *Madisen et al., 2012*), all of which had intact binocular vision (*Figure 5A*, left column; control n = 948 trials in four mice, PV-ChR2 n = 911 trials in four mice). On a third of trials, light was delivered through the implanted optic fibers during the decision period (470 nm, 5 mW/mm$^2$, 40 Hz, 50% duty cycle). Control animals showed no change in performance with the laser on (ANOVA; laser off vs. on, failure *F* = 0.030, p=0.864, success *F* = 0.026, p=0.872, abort *F* = 0.070, p=0.793), whereas PV-ChR2 animals (see *Video 4*) showed a significant reduction in performance across distances (*Figure 5B*; ANOVA; laser off vs. on, failure *F* = 7.836, p=0.008, success *F* = 15.252, p=3.35e-4, abort *F* = 10.876, p=0.002; see *Figure 5—figure supplement 1* for a breakdown of all three outcomes). On success trials, the mean landing position of PV-ChR2 mice was significantly changed compared to controls (*Figure 5C*; *t*-test, PV-ChR2 vs. control, p=0.019) while the standard deviation of landing position was not significantly different (*t*-test, PV-ChR2 vs. control, p=0.249). Interestingly, similar to mice that underwent monocular occlusion, PV-ChR2 mice showed decreased head pitch on success trials with the laser on (*Figure 5D*; *t*-test, PV-ChR2 vs. control, p=0.004), with no change in the standard deviation of pitch (*t*-test, PV-ChR2 vs. control, p=0.268) or in yaw (*t*-test, PV-ChR2 vs. control, mean p=0.116, SD p=0.164). There was no laser-associated change in trial duration, movement cluster frequency, or amplitude (data not shown; *t*-test, trial duration p=0.391; ANOVA, frequency p=0.106, amplitude p=0.930). Together, these data show that suppression of binocular zone V1 in animals with intact binocular vision significantly decreases distance estimation task performance, and that the ability to successfully perform the task is associated with changes in both pre-jump behavior and landing position.

We next asked whether mice with monocular vision would be affected by binocular zone V1 suppression (*Figure 5A*, middle column). Interestingly, PV-ChR2 mice showed no change in success rate (*Figure 5B*; ANOVA; laser off vs. on, failure *F* = 2.388, p=0.130, success *F* = 0.389, p=0.536, abort *F* = 0.090, p=0.766), and no change in landing position (*Figure 5C*; *t*-test; PV-ChR2 vs. control, mean p=0.189, SD p=0.955) or head position (*Figure 5D and E*; *t*-test; PV-ChR2 vs. control, pitch mean p=0.153, pitch SD p=0.117, yaw mean p=0.903, yaw SD p=0.849). There were also no laser-associated changes in trial duration, movement cluster frequency, or amplitude (data not shown; *t*-test, trial duration p=0.712; ANOVA, frequency p=0.882, amplitude p=0.296). This suggests that once animals switch to using monocular cues, binocular zone V1 is no longer required for accurate task performance, and that therefore the peripheral visual field is engaged.

Lastly, we asked whether suppression of monocular zone V1, which corresponds retinotopically to the peripheral visual field, affected task performance of mice with monocular vision (*Figure 5A*, right column). PV-ChR2 mice were less successful at the task with optogenetic suppression due to an increase in the number of abort trials, whereas the fraction of failure trials remained unchanged (*Figure 5B*; ANOVA; laser off vs. on, failure *F* = 0.555, p=0.462, success *F* = 10.120, p=0.003, abort *F* = 4.663, p=0.039). On success trials, PV-ChR2 mice showed no significant change in landing position (*Figure 5C*; *t*-test; PV-ChR2 vs. control, mean p=0.189, SD p=0.200); however, they showed changes in both pitch mean (*Figure 5D*; *t*-test; PV-ChR2 vs. control, mean p=0.002, SD p=0.486) and yaw standard deviation (*Figure 5E*; *t*-test; PV-ChR2 vs. control, mean p=0.388, SD p=0.018), consistent with

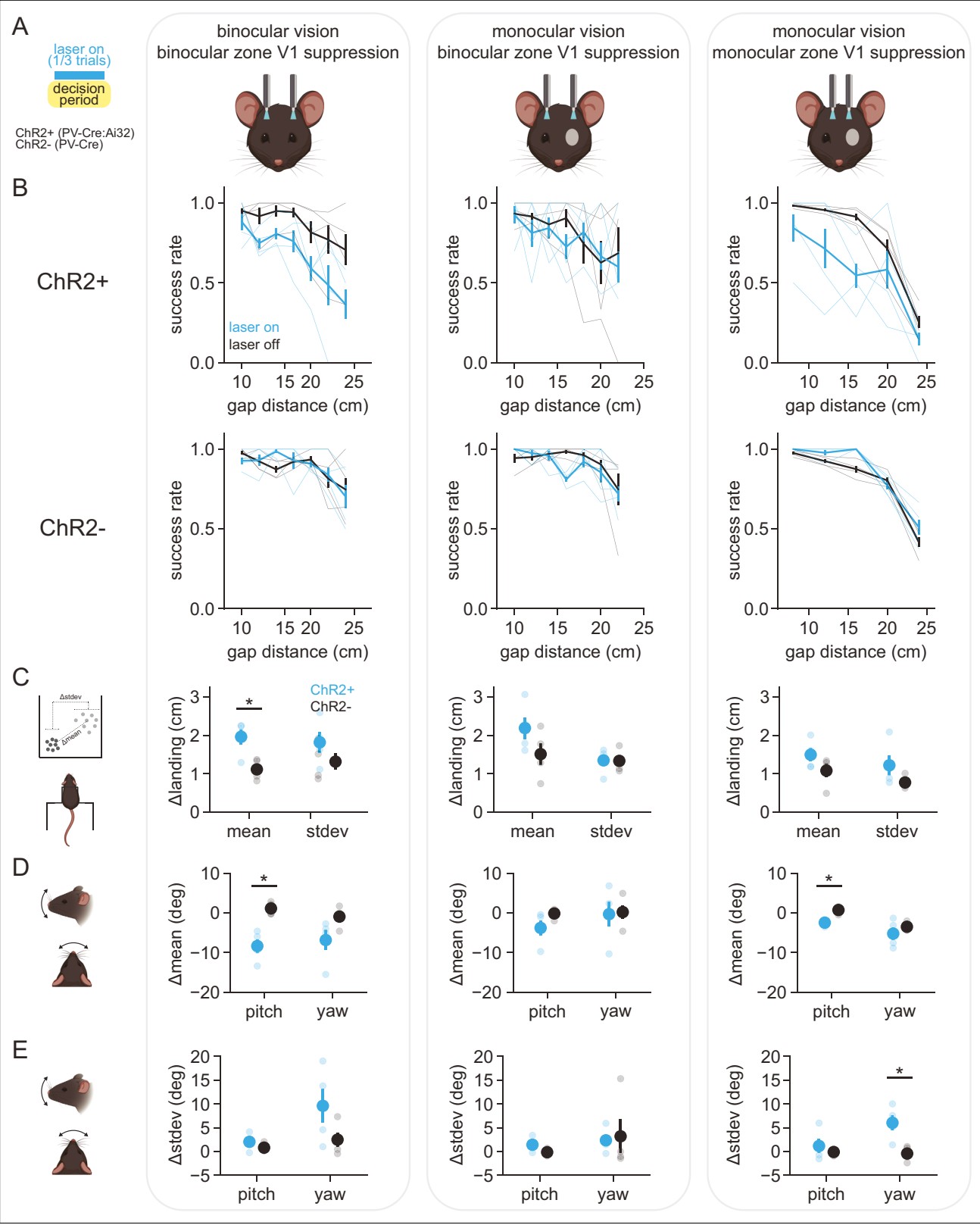

**Figure 5.** V1 optogenetic suppression disrupts distance estimation task performance. (**A**) Schematic of experimental setup for optogenetic experiments; bilateral illumination of either binocular or monocular zone V1 in either binocular or monocular animals during the decision period on one-third of trials. All plots within a column correspond to the schematized condition. (**B**) Performance curves for laser-off (black) and laser-on (cyan) conditions in mice expressing ChR2 in PV+ inhibitory interneurons (ChR2+, top row) or PV-Cre only mice (ChR2−, bottom row). Thin lines are individual animal data.

*Figure 5 continued on next page*

*Figure 5 continued*

(**C**) Change in the mean and standard deviation of landing positions, averaged across mice. Small circles are individual animal data. (**D**) Change in the mean head angle for up-down (pitch) and side-to-side (yaw) head position. (**E**) Same as (**D**) but change in standard deviation of pitch and yaw.

The online version of this article includes the following figure supplement(s) for figure 5:

**Figure supplement 1.** V1 optogenetic suppression task performance.

shifting head position to utilize a portion of the visual field unaffected by optogenetic suppression. There was no laser-associated change in trial duration, movement cluster frequency, or amplitude (data not shown; *t*-test, trial duration p=0.050; ANOVA, frequency p=0.476, amplitude p=0.344). It should be noted these experiments were performed in a separate group of mice with narrower landing platforms, thus comparisons of success rate and landing position to the other experimental groups may not be insightful.

In summary, these experiments show that manipulating V1 activity significantly alters the behavior of mice on the distance estimation task, and the correspondence between the anatomical locus of suppression and the ocular condition supports a role for V1 in both binocular and monocular cue-mediated distance estimation task performance.

## Discussion

We have established a visual distance estimation task in mice that engages an ethological, freely moving behavior. Previous research using similar versions of this task suggests that gerbils and rats utilize multiple cues to determine the distance to objects in the environment, including retinal image size, binocular vision, and motion parallax (*Carey et al., 1990*; *Ellard et al., 1984*; *Goodale et al., 1990*; *Legg and Lambert, 1990*). Importantly, this task is distinct from 'gap-crossing' tasks (*Hutson and Masterton, 1986*) where animals can use the whisker somatosensory system to determine the distance across a short gap. Furthermore, in contrast to other recently developed tasks that are designed to probe binocular depth perception and stereopsis (*Boone et al., 2021*), in this task mice are able to use monocular cues for depth, including those that are generated by self-movement. It can therefore be flexibly used to investigate a variety of distance estimation tactics by manipulating experimental conditions.

### Cues for distance estimation

Binocular vision (and therefore stereopsis) was not required for accurate distance estimation in this task, consistent with previous studies on gerbils (*Ellard et al., 1984*). This provides the first demonstration that mice are able to use depth cues that are available besides stereopsis. Vertical head movements sufficient to generate motion parallax cues are increased in frequency under monocular conditions, suggesting that mice may use motion parallax under monocular vision in this task. Previous work found increasing frequency of vertical head movements as a function of gap distance in gerbils and rats (*Ellard et al., 1984*; *Legg and Lambert, 1990*). We did not see such a relationship, which could be due to species-specific differences or differences in task design. Interestingly, there was an inverse relationship between changes in movement frequency and head angle – animals with the smallest change in frequency of head movements showed the largest change in downward head angle. This could reflect the use of position parallax cues (comparing two perspectives, initial and final, rather than motion cues) and was also present in animals with binocular vision when binocular zone V1 activity was suppressed. These results do not rule out the use of binocular disparity under normal conditions – in fact, changes in decision period head position and movement between binocular and monocular conditions, in addition to our optogenetic

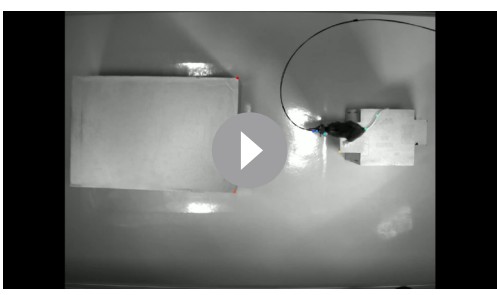

**Video 4.** PV-ChR2 mouse with binocular vision during a laser-off and a laser-on trial for binocular zone V1 suppression.
https://elifesciences.org/articles/74708/figures#video4

suppression experiments in binocular zone V1, provide evidence that animals use binocular cues as well under normal conditions.

The increase in decision period duration and total distance moved by mice in the monocular relative to binocular condition is consistent with an active sensing strategy that requires temporal integration. The inherent separation of the decision period from the jump execution in this task helps separate potential active sensing behavior from pure jumping behavior – in theory, the animal need not make any movements except those required to jump (*Stamper et al., 2019*). Closed-loop control of the sensory environment is a powerful tool for investigating active sensing (*Biswas et al., 2018*) in this task, altering the landing platform based on head movements would provide a causal test to determine whether mice use motion parallax cues, as was performed in locusts (*Wallace, 1959*). Furthermore, investigating the relationship between active sensing and memory-guided visual behavior could provide new insights into natural behavior and its neural basis. For example, experiments with a modified version of this task suggest that parietal cortex is necessary for context-dependent use of retinal image size cues (*Ellard and Sharma, 1996*). Future research could investigate how these contextual associations are formed through active sensing and bound into memory in the brain.

## Eye movements during distance estimation

Using miniature cameras to track the eyes, we found that eye movements compensate for head movements to stabilize gaze leading up to the jump. This would allow the mouse to both maintain gaze toward the platform and reduce motion blur throughout large-amplitude head movements (*Land, 1999*). This is consistent with previous work showing that mouse eye movements stabilize gaze during both operant behavior (*Meyer et al., 2020*) and a natural behavior, prey capture (*Michaiel et al., 2020*). It will be interesting to determine whether jumping mice control their gaze to localize the platform on a specific subregion of the retina; that is, whether there exists a retinal specialization for determining distance. In the case of prey capture, the image of the cricket is stabilized in the retinal region with the highest concentration of alpha-ganglion cells (*Holmgren et al., 2021*). We also found that mice did not move their eyes to increase binocular overlap during the period immediately preceding the jump, similar to a previous finding demonstrating that rats do not align the gaze of the two eyes before crossing a short gap (*Wallace et al., 2013*). Smooth eye movements provide extra-retinal signals for computing depth from motion parallax in primates (*Kim et al., 2017*; *Nadler et al., 2008*; *Nadler et al., 2009*), and future studies may address whether the compensatory movements we observed play a similar role in mice. Finally, these experiments show that mice are capable of performing this task with a tether and significant hardware weight on the head, which is a critical requirement for introducing additional techniques such as electrophysiology into this paradigm.

## Neural circuits underlying distance estimation

We provide evidence for V1 specifically being important for distance estimation behavior in mice. Given the large volume of V1 (~5 mm$^3$) relative to the spread of laser-induced suppression in PV-ChR2 mice (~1 mm$^3$, *Li et al., 2019*), animals likely maintained some V1 function across all conditions, which would explain why animals in all conditions could still perform the task to some degree, and did so with associated changes in how they orient their heads (i.e., using different parts of the visual field). Whether animals would perform this task with total V1 suppression is unclear given the option to abort trials. However, the fact that the anatomical locus of suppression (binocular vs. monocular zone V1) determined whether there was an effect on behavior between ocular conditions (binocular vs. monocular vision) supports the hypothesis that mice use V1 to estimate distance in this task. In fact, binocular V1 suppression led to the same changes in decision period head angle in mice with binocular vision that monocular occlusion induced, suggesting a common shift in strategy with the loss of stereopsis.

These results are consistent with previous work showing broad lesions of occipital cortex disrupt performance without affecting head movements, whereas lesions to superior colliculus and preoptic area had no effect on either (*Ellard et al., 1986*). This task could therefore be a useful tool for studying the specific computations performed in V1 that mediate accurate distance estimation, and both the visual and nonvisual input signals required to perform these computations. Additionally, the neural circuits that convert visual information into a jump command are also not well understood. Most work has examined jumping in nocifensive and defensive contexts rather than navigation (*Barik et al., 2018*; *Wang et al., 2015*), although a recent behavioral study demonstrated that squirrels learn to

integrate multiple factors, including gap distance and branch flexibility, in executing a jump (*Hunt et al., 2021*).

## Utility of studying natural distance estimation behavior

Natural behavior is often a continuous control process, which is fundamentally closed-loop, unlike stimulus-response paradigms that dominate behavior literature (*Cisek, 1999*). This task accordingly permits investigation of both how movement through the environment generates sensory cues useful for judging distance, and how the visual information is directly converted into a motor output. Furthermore, perception of spatial layout is an embodied process, and thus body- and action-scaling cues that are not available under conditions of restraint could provide distance information under the freely moving conditions of this task (*Fajen, 2021*). Critically, natural behaviors may be the most appropriate tool for studying the neural basis of sensory processing since theoretical considerations suggest that neural circuits may perform suboptimal inference under non-natural conditions (*Beck et al., 2012*). Finally, beyond studies of visual distance estimation, this task could provide a framework for integrated studies of motivation, motor control, and decision-making within an ethological context.

# Materials and methods

## Animals

Male and female mice between postnatal day 40 (P40) and P365 were bred in-house in a C56BL/6J background. For optogenetic experiments, transgenic mice were used to target the expression of channelrhodopsin-2 to parvalbumin-expressing neurons (PV-Cre [B6;129P2-Pvalbtm1(cre)Arbr/J, Jax #008069] crossed to Ai32 [B6;129S-Gt(ROSA)26Sortm32(CAG-COP4*H134R/EYFP)Hze/J, Jax #012569]; *Hippenmeyer et al., 2005*; *Madisen et al., 2012*). Mice were housed in a reverse 12 hr light–dark cycle room. Mice were placed under a water restriction schedule at the start of training, only receiving fluids during training/task periods. All procedures were performed in accordance with the University of Oregon Institute for Animal Care and Use Committee and Animal Care Services standard operating procedures.

## Behavioral apparatus and jumping task

Two cohorts of mice were used in this study, with a subset of experiments (eye cameras and monocular V1 suppression) using a smaller arena, narrower landing platforms, and fewer gap distances. All other experimental components of the task were identical. The jumping arena was roughly 45 cm high, 70 cm wide, and 100 cm across for most experiments, and was 30 cm high and 60 cm across for eye camera/monocular V1 suppression. Mice self-initiated trials by mounting a take-off platform (15 cm height, 10 cm width, 10 cm depth, with 4 × 5 cm overhang in front). While blocking the mouse's view of the arena with a barrier, the experimenter then placed one of three platforms (15, 20, or 25 cm width, 30 cm depth, 19 cm height for most experiments; 10, 20, or 30 cm width, 5 cm depth, 19 cm height for eye camera/monocular V1 suppression) at a random distance (10, 12, 14, 16, 18, 20, or 22 cm for most experiments; 8, 12, 16, 20, or 24 cm for eye camera/monocular V1 suppression) from the edge of the take-off platform. Platforms were custom built from ¼" acrylic, and tops were coated in white rubberized coating (Plasti-Dip) or fine-grained white sandpaper to prevent animals from slipping. For eye camera/monocular V1 experiments, the platforms were white and a black strip was placed across the top leading edge of the landing platform, matched proportionally in height to platform width to maintain height/width ratio. Arena and platforms were constructed by the University of Oregon Technical Science Administration. A static white noise background composed of grayscale squares (~1° each of visual angle from take-off platform) was mounted at the back of the arena. Six LED puck lights were evenly spaced around the top of the arena for even illumination. Cameras (FLIR BlackFly S USB3) were mounted above and to the side of the arena, and the entire behavioral session was recorded (1440 × 1080 pixels at 99.97 fps, or 720 × 540 pixels at 60 fps) with camera timestamps using a custom Bonsai workflow (*Lopes et al., 2015*). A custom Python script was used to generate randomized platform/distance combinations for the experimenter and to log trial outcomes and approximate jump times. The moment the barrier was lifted and the mouse was able to see the landing platform constituted the trial start, and the time elapsed until the mouse jumped was the 'decision period.' There were three possible trial outcomes: (1) the mouse jumped and successfully

reached the landing platform and received a reward (success), (2) the mouse jumped and missed the landing platform and received no reward (failure), or (3) the mouse dismounted the take-off platform and received a light airpuff and a time-out (abort).

All experiments included data from eight mice (same mice for binocular, monocular, and binocular V1 suppression; same mice for eye cameras, monocular V1 suppression, and supplementary ARHMM figure) and typically lasted 30–45 min. The mean ± standard error for the number of sessions and trials per mouse was as follows: binocular sessions 11.4 ± 0.6, trials 42.1 ± 2.3; monocular sessions 5.6 ± 0.2, trials 39.0 ± 3.4; eye cameras sessions 2.8 ± 0.3, trials 13.9 ± 0.7; binocular vision/binocular zone V1 suppression sessions 6.8 ± 0.2, trials 34.3 ± 2.6; monocular vision/binocular zone V1 suppression sessions 4.8 ± 0.2, trials 33.5 ± 1.8; monocular vision/monocular zone V1 suppression sessions 3.6 ± 0.2, trials 27.2 ± 1.7. The number of trials was partly limited by the manual nature of the task, which required an experimenter to manually place a platform at a specific distance. However, this process typically took less than 3 s, whereas the time spent consuming the reward and freely investigating the arena between trials was significantly longer. Future versions of the task with automated platform placement and reward delivery could increase the number of trials per session, and increase the length of individual sessions. Acquisition files are available online at https://github.com/nielllab/nlab-behavior/tree/master/jumping/bonsai(copy archived at swh:1:rev:44a40fdbd63ed7740a73b8d-085333c8d1b22c592; path=/jumping/bonsai/; *niellab, 2022*).

## Behavioral training

Mice were habituated to the arena for 3 days with their cage mates, during which time they were individually handled and introduced to a clicker that indicated water reward (~25–50 ul), where each click is immediately followed by a reward. Mice were then individually clicker trained to mount a short take-off platform (10 cm height; click and reward upon mounting the platform), receiving water (administered by hand using a 1 ml syringe), and a small piece of tortilla chip (Juanita's). After 3–5 successful mounts, a landing platform (19 cm height) was placed against the take-off platform, and mice were clicker-rewarded for climbing up onto the landing platform. After three successful trials, the landing platform was moved slightly farther back, increasing the gap distance until jumping is required to reach the landing platform. At this point, the clicker was typically no longer required. Once the mouse could jump to the maximum distance, the taller take-off platform used in the task was introduced, and landing platforms were again introduced at short distances and slowly moved farther away. Training was complete when mice could jump to all three landing platforms at the farthest distance, and typically took 1–2 weeks with all mice successfully learning the task.

## Surgical procedures

For all procedures, anesthesia was induced at 3% isoflurane and maintained at 1.5–2% in $O_2$ at a 1 l/min flow rate. Ophthalmic ointment was applied to both eyes, and body temperature was maintained using a closed-loop heating pad at 37°C. In order to minimize stress when plugging in optical tethers or miniature cameras, a small steel headplate was mounted on the skull using dental acrylic (Unifast LC) to allow for brief head-fixation before the experiment.

## Monocular suture

The area immediately surrounding the eye to be sutured was wiped with 70% ethanol before ophthalmic ointment was applied. Two to three mattress sutures were placed using 6-0 silk suture, opposing the full extent of the lid. The forepaw and hindpaw nails ipsilateral to the sutured eye were trimmed to help minimize postprocedural self-inflicted trauma.

## Optic fiber implant

A minimal portion of scalp was resected bilaterally over visual cortex, and a small trepanation was made over each primary visual cortex (+1.0, ±2.5 mm for monocular zone or +1.0, ±3.0 mm for binocular zone, relative to lambda suture). Bilateral optic fibers (ceramic ferrules, thorlab fiber 0.5 mm length from end of ferrule) were stereotactically lowered into the burr hole and secured in place with dental acrylic. Vetbond was then applied to secure the skin in place around the implant. Fiber transmission rates were measured prior to implant and accounted for during experiments.

## Miniature head-mounted cameras

To obtain high-resolution video of the eyes during behavior, a miniature camera (iSecurity), magnifying lens (12 mm focus, 6 mm diameter), and an infrared LED were mounted on a custom-designed 3D-printed plastic camera arm (*Michaiel et al., 2020*). Two miniature connectors (Mill Max 853-93-100-10-001000, cut to 2 × 4 pin) were glued to the headplate, and an equivalent connector on the camera arm was plugged in prior to the experiment. Camera power and data were passed through thin tethering wire (Cooner #CZ1174CLEAR) and acquired in Bonsai with system timestamps. The total hardware weight was approximately 3 grams. Eye videos were deinterlaced to achieve 60 fps (matching top/side cameras) prior to analysis.

## Data analysis

Full task videos were first split into individual trials using custom Python software; the trial start, jump, and landing frame numbers were determined and individual trial videos were saved. These trial videos were then labeled using markerless pose estimation with DLC. A set of sample frames were manually labeled and used to train two networks (top/side cameras and eye cameras) that were then used to track features in all video data. The distance jumped was calculated using the position of the left ear in the top-camera frame where the front paw touched the platform and the animal decelerated (success trials) or when the front paw passed below the edge of the landing platform (failure trials). DLC points from the side-camera data during the decision period were passed through a median filter (n = 3) and a convolutional smoothing filter (box, n = 5). To extract individual movements, we identified all zero crossings in the velocity trace calculated from eye position, then extracted the 500 ms period around those time points in the eye position data. Movements that overlapped by more than 250 ms with a previous movement were excluded from further analysis. The x and y values were concatenated, and all movements across all conditions were fed into principal component analysis, after which k-means clustering was performed on the reduced data. We tried varying the number of clusters across a range of values (4–20) and found that k = 10 resulted in clusters that appeared to contain a single type of movement while minimizing repeated clusters of the same movement type. Pitch and yaw were both calculated from the angle between the eye and ear in the side and top video data, respectively.

For hidden Markov model (ARHMM) analysis, values of the points tracking the nose, eye, and ear were used as inputs for training after centering across experiments by subtracting off values of the point that tracked the edge of the take-off platform. Model training was performed using the SSM package in Python (*Linderman et al., 2019*). Model selection was based on the elbow in twofold cross-validation log-likelihood curves across model iterations while balancing model interpretability with model fit (final model: K = 6, lag = 1, kappa = 1e04, data temporally downsampled 2×, and one state discarded due to extremely low prevalence). ARHMM states were determined based on a posterior probability threshold of 0.8. Time points below the threshold were excluded from analysis. For lexical transition matrices, trials were first separated into binocular and monocular conditions. During the decision period of each trial, the transitions between unique ARHMM states were counted. The number of state transitions was then normalized by the total number of unique transitions per condition to calculate the relative frequency of transitions. For all summary analyses, data were first averaged within animal (across days) and then across animals within a group (e.g., monocular, binocular). Statistical significance was determined using ANOVA and the Student's *t*-test with Bonferroni corrections for multiple comparisons. Unless otherwise noted, data are presented as mean ± standard error of the mean. Analysis code is available online at https://github.com/nielllab/nlab-behavior/tree/master/jumping/python%20files (copy archived at swh:1:rev:44a40fdbd63ed-7740a73b8d085333c8d1b22c592; path=/jumping/python%20files/; *niellab, 2022*). Data are available at https://doi.org/10.5061/dryad.r7sqv9sg2.

## Decoding analysis

We decoded the experimental condition (binocular vs. monocular, laser on vs. off) per animal from single-trial maximum a priori (MAP) motif sequences inferred using the ARHMM. Specifically, we trained binary decoders with a linear decision boundary (linear discriminant analysis) to decode the above categorical variables from the single-trial empirical state transition probability matrices derived from the MAP sequence of each trial, thus providing not only state usage information, but transitions between states as information the classifier could use. For each animal, correct trials

were pooled across distances to provide enough trials per class for decoding. Data were split into training and test datasets in a stratified 10-fold cross-validation manner, ensuring equal proportions of trials of different types (distance, platform width, visual condition, laser) in both datasets. To calculate the statistical significance of decoding accuracies, we performed an iterative shuffle procedure on each fold of the cross-validation, shuffling training labels and testing on unshuffled test labels 100 times to create a shuffle distribution for each fold of the cross-validation. From these distributions, we calculated the z-score of decoding accuracy for each class in each cross-validation fold. These z-scores were then averaged across the folds of cross-validation and used to calculate the overall p-value of the decoding accuracy obtained on the original data. The decoding weights of the binary classifiers were examined as well to identify the significant transitions that contributed to decoding between visual conditions. The same shuffle procedure was used to assess significant elements of the classifier.

## Statistics

All summary data in text and plots, unless noted otherwise, are mean ± standard error. All statistical tests were performed with SciPy, and in cases where data were not normally distributed, nonparametric tests were used.

## Acknowledgements

We thank members of the Niell lab for helpful discussions, and Dr. Michael Goard, Dr. David Leopold, Dr. Matt Smear, and Dr. Michael Stryker for feedback on the manuscript. This work benefited from access to the University of Oregon high-performance computer, Talapas, and the University of Oregon Technical Science Administration. This work was supported by grants from the National Institutes of Health: 5F32EY027696 (Parker), 1R21EY032708 (Niell, Parker), 1R34NS111669 (Niell). Some figure panels were made using https://biorender.com/.

## Additional information

### Funding

| Funder | Grant reference number | Author |
| --- | --- | --- |
| National Institutes of Health | 5F32EY027696 | Philip RL Parker |
| National Institutes of Health | 1R21EY032708 | Philip RL Parker |
| National Institutes of Health | 1R34NS111669 | Cristopher M Niell |

The funders had no role in study design, data collection and interpretation, or the decision to submit the work for publication.

### Author contributions

Philip RL Parker, Conceptualization, Data curation, Formal analysis, Supervision, Funding acquisition, Writing – original draft, Writing – review and editing; Elliott TT Abe, Software, Formal analysis, Visualization, Methodology; Natalie T Beatie, Data curation, Investigation; Emmalyn SP Leonard, Data curation, Investigation, Methodology; Dylan M Martins, Data curation, Software, Formal analysis; Shelby L Sharp, Investigation; David G Wyrick, Software, Formal analysis, Investigation; Luca Mazzucato, Formal analysis, Supervision, Methodology; Cristopher M Niell, Conceptualization, Resources, Supervision, Funding acquisition, Project administration, Writing – review and editing

### Author ORCIDs

Philip RL Parker  http://orcid.org/0000-0002-6224-9747
Luca Mazzucato  http://orcid.org/0000-0002-8525-7539
Cristopher M Niell  http://orcid.org/0000-0001-6283-3540

## Ethics

This study was performed in strict accordance with the recommendations in the Guide for the Care and Use of Laboratory Animals of the National Institutes of Health. All procedures were performed in accordance with the University of Oregon Institute for Animal Care and Use Committee and Animal Care Services standard operating procedures, under approved protocol AUP-21-21.

## Decision letter and Author response

Decision letter https://doi.org/10.7554/eLife.74708.sa1
Author response https://doi.org/10.7554/eLife.74708.sa2

## Additional files

### Supplementary files
• Transparent reporting form

### Data availability
Data are available at https://doi.org/10.5061/dryad.r7sqv9sg2.

The following dataset was generated:

| Author(s) | Year | Dataset title | Dataset URL | Database and Identifier |
|---|---|---|---|---|
| Parker PR | 2022 | Distance estimation from monocular cues in an ethological visuomotor task | https://doi.org/10.5061/dryad.r7sqv9sg2 | Dryad Digital Repository, 10.5061/dryad.r7sqv9sg2 |

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
