## [Editor Report]

This is an important article with compelling experimental methods, including ethologically relevant behavior, sophisticated physiological methods including optogenetic suppression of primary visual cortical activity, careful behavioral experiments, and clear, convincing, quantitative analysis of the resulting data. The article enhances our understanding of the role of active visual estimation of distance under multiple factors of visual degradation (binocular/monocular, and V1 suppression), demonstrating how robust task performance can emerge from compensatory active sensing strategies.

---

## [Decision Letter]

**Decision letter after peer review:**

Thank you for submitting your article "Distance estimation from monocular cues in an ethological visuomotor task" for consideration by *eLife*. Your article has been reviewed by 3 peer reviewers, including Noah J Cowan as Reviewing Editor and Reviewer #1, and the evaluation has been overseen by Tirin Moore as the Senior Editor.

Essential revisions:

The reviewers all agree that the manuscript has potential for publication, and that the data are sound, but there was broad and unanimous agreement that the paper needs significantly stronger analysis of the data. While one reviewer mentioned that neural recordings would broaden the appeal, during consultation it was agreed that such recordings were beyond the scope of this work. In short, it was agreed that deeper analysis of the behavioral data is needed and will strengthen this paper, and that neural recordings or other additional data are not necessary.

The level of effort and care that went into the experiment design, experiments, and data analysis is highly commendable. This treasure trove of data, as currently analyzed, appears to establish these basic results:

– Monocular cues are *sufficient* for jumping nearly equally well.

– There is some sort of increased activity before jumping in the monocular setting.

– V1 plays some general role.

– Eye movements are anti-correlated with head movements (a broadly known idea across taxa and completely expected) and do not seem to increase binocular overlap (much more relevant to the current task).

All three reviewers, using different language, felt that the analysis of the pre-jump behavior was incomplete, not well illustrated, and unclear. It is evident that a more thoughtful and deeper dive into the behavioral data is essential to be considered for publication. Relating these data, for example, to gaining additional disparity or other possible behavioral mechanisms for gaining sensory information for the decision making task would greatly enhance the manuscript.

Likewise, the analysis of the V1 data just scratches the surface. The authors simply noted that there were more aborts, but provided no mechanistic insights, and the manuscript failed to firmly establish that V1 is used in the distance estimation component.

The reviewers all raised individual questions that should be addressed in a response document in a point-by-point manner, if the authors choose to revise the manuscript for resubmission.

*Reviewer #1 (Recommendations for the authors):*

As mentioned above in the public review, there are two major weaknesses with the HMM-only approach to data analysis of the binocular vs. monocular behavior. First, it would seem that if there are major differences in pre-jump behavior, that much simpler statistical measures (root-mean-squared velocities of the head, for instance?) that should pull out the differences in active sensing, and such "simple" analysis which not be as obfuscating. If that fails, then the measures by which you tried this and they failed should be reported, and the fancy HMMs should be thusly motivated. Second, no mechanistic insight was garnered. What is the nature of the additional active sensation? Is it larger amplitude, for instance, so as to gain better disparity? Or is it just occurring over a longer period of time, suggesting a temporal integration strategy?

However, maybe a deeper dive into the data could reveal more. Rather than simply looking at success / fail, would it be possible to examine jumping distance in the various conditions? Since the authors used DLC, they have metric information about jumping that does not seem to be getting used. Did the V1-inhibited mice have a wider variance on their jumping distance when they did jump, for example? Such analyses would be more fine grained and might pick up subtle differences that the binary "success / abort / fail" cannot, in order to better quantify the importance of V1 in distance estimation vs. cognitive decision making about gap jumping. It was also unclear what the authors meant when they said the HMM's concluded there was no change in behavior – was this only measured for cases where there was an attempted jump? Again, the HMMs really obscure these results and it would be good to check other simpler movement statistics to motivate and/or confirm the HMM results.

The authors state that the V1-inhibited group showed little-to-no change failure rate (which I didn't not see defined, but I assume this means the proportion of failures compared to jumping attempts). This suggests that, as an alternative to the authors' primary claim that distance judgements are impaired, perhaps the *decision* to make a jump is inhibited, but not necessarily the distance estimation itself. I'm not enough of an expert in the various visual pathways, but it seems like this paper might support the view that distance judgements for the purpose of decision making may use different mechanisms and/or pathways than distance estimation during execution. Can the authors clarify this?

The gaze stabilization results are interesting, but it seems that the main point is this statement: "demonstrating that mice do not move their eyes to increase binocular overlap preceding the jump." In the discussion, I suggest this be brought up again, I think, in the section "Cues for distance estimation". After the authors state "This provides the first demonstration that mice are able to use depth cues that are available besides stereopsis." it would be good to point out that the eyes do not appear to be attempting to improve stereopsis based on eye movement patterns, and that instead the results are consistent with the fact that binocular vision may be used in a standard sensory fusion kind of way, i.e. two measurements simply reduce uncertainty. The animals compensate for this via more active sensing and then achieve similar performance. (I really appreciate the reference to Wallace (1959) and hope this experiment is one-day performed in mice!) To summarize my point in this paragraph, could it be that the evidence from this paper supports that, at least for this task, binocular cue integration in mice is just "extra sensing" i.e. the use of two eyes is the "sum of the parts" are not "stereoscopic vision" i.e. "greater than the sum of the parts"?

Finally, I feel there is a wide range of active sensing literature that the authors could include in a discussion, since ultimately the most compelling potential finding in this paper is the increase in active sensing as a function of loss of binocular vision. As a starting point you could check out our recent papers for pretty extensive bibliographies on active sensing (no need to cite our papers unless you deem some aspect as appropriate):

S. A. Stamper, M. S. Madhav, N. J. Cowan, and E. S. Fortune, "Using Control Theory to Characterize Active Sensing in Weakly Electric Fishes," in Electroreception: Fundamental Insights from Comparative Approaches, B. Carlson, J. Sisneros, A. Popper, and R. Fay, Eds., New York: Springer Science+Business Media, LLC, 2019, vol. 70.

D. Biswas, L. A. Arend, S. Stamper, B. P. Vágvölgyi, E. S. Fortune, and N. J. Cowan, "Closed-Loop Control of Active Sensing Movements Regulates Sensory Slip," Curr Biol, vol. 28, iss. 4, 2018.

Perhaps the best connection of any to our work is the increase in active sensing with the loss of a sensory system – in this case perhaps the loss of a single eye would be analogous to the loss of a modality that causes increases to active sensing in a different modality. The change in active sensing strategy here is similar but suggests that the partial loss of a modality can lead to increases in active sensing in the same modality. Again, this connection is not critically important, just an observation that the authors may or may not find relevant to their discussion.

*Reviewer #2 (Recommendations for the authors):*

1. The animal's success rate drops with gap distance. Is that caused by errors in vision (distance estimation) or jumping (motor accuracy). I wonder if the authors have data to address this question.

2. It is true that mice can use monocular cues to estimate distance in this task, but the observed differences between monocular and binocular conditions (more head movements and the difference in the modeling results) actually suggest that mice normally use binocular cues. The authors should state this conclusion more prominently.

3. Related to the point #2 above, it would be interesting to know if the performance requires visual cortex under binocular condition.

4. The performance was impaired upon suppressing the visual cortex, but the animals can still do it. The authors discussed this as incomplete suppression. This is reasonable, but are there data to support this, like recording across the cortical depth and visual cortex? Related, the suppression was considered "V1", but how did the authors confirm that this was restricted to V1, and how much of V1 was illuminated by the light?

*Reviewer #3 (Recommendations for the authors):*

Since the development of a new task is a major part of the manuscript, I recommend more details in the Materials and methods for describing the task. Based on the current methods section, I assume the platform and distance were manually changed by the experimenter for each trial. Please provide more detail on the time it took for this manual step and discuss how this might or might not affect the animals' engagement in the task. Were the animals conditioned to this experimenter intervention? In addition, this labor-intensive step limits the number of trials mice can perform (30-60 per session). There should be information regarding the number of sessions each mouse performed. Related, I would like to see the behavior and optogenetics data from each animal plotted separately.

If the claim of mice accurately estimated the distance was supported by the bottom panel of B and C on Figure 2, then please show both the jump distance for both the success trials and failed trials. Do mice often fall short on the failed trials compared to the success trials?

What is the image inside of each circle in figure 3A? Is it a mouse? I am unclear how it is related to each movement state of the hidden Markov model. Please consider more informative illustrations. Related, I find the video showing the Deeplabcut markers used very helpful, please consider having a panel in figure 3 showing the markers feed into the hidden Markov model. This would also help make the example eye position traces in panel A clearer.

The font of the numbers inside the matrix on figure 3B is too small, please make them bigger or remove them.

Most importantly, the conclusion from Figure 3 is Mice perform more head movements under monocular conditions. This conclusion wasn't intuitive from more transitions between states in the monocular condition. It is clear from the example traces that the states represent different movements, but more transition in states being the equivalent of more head movement would at least require example traces of head position around the transition in states.

How many animals were used in the eye tracking experiments? The jump distance in Figure 4 panel C seems to be lower than the jump distance in Figure 2B and C, especially for the lower gap distance. Is this the strategy of the animals used in eye tracking, or is the eye tracking equipment weighing down the animals? Showing the success rate and jump distance of individual animals would be helpful.

As mentioned in the text, the change in performance during optogenetic suppression trials was largely due to an increase in abort with little change in failure rate. Therefore, showing an example video of an animal that failed a trial might be misleading.

Reference in the middle of the paragraph at Discussion, Eye movements during distance estimation section. Holmgren et al., is missing a date.

---

## [Author Response]

Reviewer #1 (Recommendations for the authors):As mentioned above in the public review, there are two major weaknesses with the HMM-only approach to data analysis of the binocular vs. monocular behavior. First, it would seem that if there are major differences in pre-jump behavior, that much simpler statistical measures (root-mean-squared velocities of the head, for instance?) that should pull out the differences in active sensing, and such "simple" analysis which not be as obfuscating. If that fails, then the measures by which you tried this and they failed should be reported, and the fancy HMMs should be thusly motivated. Second, no mechanistic insight was garnered. What is the nature of the additional active sensation? Is it larger amplitude, for instance, so as to gain better disparity? Or is it just occurring over a longer period of time, suggesting a temporal integration strategy?

We have made significant changes to the analyses to incorporate simpler and more interpretable measures of behavior. The simplest analysis is the duration of the trials and the distance traveled during that period. Animals in the monocular condition showed significantly longer decision periods with more overall movement, consistent with a temporal integration strategy. Next, in order to make the individual movements more intuitive, we identified movements using zero-velocity crossings of x/y eye position, and clustering with PCA/k-means, and found that the amplitude of individual movements was unchanged between binocular and monocular conditions, while frequency was increased. Finally, we analyzed the angle of the head, which showed a systematic within-animal change between binocular and monocular conditions. Together, we feel these findings (Figure 3) provide more mechanistic insight into the decision strategy.

However, maybe a deeper dive into the data could reveal more. Rather than simply looking at success / fail, would it be possible to examine jumping distance in the various conditions? Since the authors used DLC, they have metric information about jumping that does not seem to be getting used. Did the V1-inhibited mice have a wider variance on their jumping distance when they did jump, for example? Such analyses would be more fine grained and might pick up subtle differences that the binary "success / abort / fail" cannot, in order to better quantify the importance of V1 in distance estimation vs. cognitive decision making about gap jumping. It was also unclear what the authors meant when they said the HMM's concluded there was no change in behavior – was this only measured for cases where there was an attempted jump? Again, the HMMs really obscure these results and it would be good to check other simpler movement statistics to motivate and/or confirm the HMM results.

In order to provide greater insight into the various conditions, we have performed further experiments and analysis. We collected additional data for optogenetic experiments to include binocular/monocular vision with binocular zone V1 suppression, and analyzed both landing position and the positioning of the head during the decision period (Figure 5). We found changes in landing position and head angle (similar to monocular suture behavior) in animals with binocular vision and binocular zone V1 suppression, and found changes in head position (including more variance in side-to-side head angle) for monocular vision and monocular zone V1 suppression. However, there was not a significant change in the variance of landing position, and head movements (using our new analysis) were unchanged even when including abort trials. These new data suggest our suppression is fairly localized relative to the size of V1, so animals are able to change their head position to perform the task, and this is specific to binocular vision/V1 or monocular vision/V1. This finding in itself supports the notion that V1 plays a role in the behavior. We have updated Figure 5 to incorporate these new data, and added relevant discussion.

The authors state that the V1-inhibited group showed little-to-no change failure rate (which I didn't not see defined, but I assume this means the proportion of failures compared to jumping attempts). This suggests that, as an alternative to the authors' primary claim that distance judgements are impaired, perhaps the decision to make a jump is inhibited, but not necessarily the distance estimation itself. I'm not enough of an expert in the various visual pathways, but it seems like this paper might support the view that distance judgements for the purpose of decision making may use different mechanisms and/or pathways than distance estimation during execution. Can the authors clarify this?

It is true that the increase in the number of aborts without a change in the number of failures makes it unclear whether distance estimation or the decision to make a jump is affected – we have changed the text to clarify that V1 suppression disrupts task performance, rather than distance estimation per se.

The gaze stabilization results are interesting, but it seems that the main point is this statement: "demonstrating that mice do not move their eyes to increase binocular overlap preceding the jump." In the discussion, I suggest this be brought up again, I think, in the section "Cues for distance estimation". After the authors state "This provides the first demonstration that mice are able to use depth cues that are available besides stereopsis." it would be good to point out that the eyes do not appear to be attempting to improve stereopsis based on eye movement patterns, and that instead the results are consistent with the fact that binocular vision may be used in a standard sensory fusion kind of way, i.e. two measurements simply reduce uncertainty. The animals compensate for this via more active sensing and then achieve similar performance. (I really appreciate the reference to Wallace (1959) and hope this experiment is one-day performed in mice!) To summarize my point in this paragraph, could it be that the evidence from this paper supports that, at least for this task, binocular cue integration in mice is just "extra sensing" i.e. the use of two eyes is the "sum of the parts" are not "stereoscopic vision" i.e. "greater than the sum of the parts"?

First, regarding whether the eyes are moving to improve stereopsis, mice have a ~40 deg zone of binocular overlap at baseline, so it does not seem likely that increasing binocular overlap would improve stereopsis, perhaps explaining why we do not see a change. Regarding whether binocular cue integration in mice is the “sum of the parts” or “extra sensing,” we cannot speak to this issue with the current dataset. However, we presume that mice use all of the cues they have available, which includes stereopsis under binocular conditions. Furthermore, analysis from our new experiments show that decision period behavior changes under monocular conditions, suggesting animals are relying on different cues under the two conditions. We have updated the discussion to include these points.

Finally, I feel there is a wide range of active sensing literature that the authors could include in a discussion, since ultimately the most compelling potential finding in this paper is the increase in active sensing as a function of loss of binocular vision. As a starting point you could check out our recent papers for pretty extensive bibliographies on active sensing (no need to cite our papers unless you deem some aspect as appropriate):S. A. Stamper, M. S. Madhav, N. J. Cowan, and E. S. Fortune, "Using Control Theory to Characterize Active Sensing in Weakly Electric Fishes," in Electroreception: Fundamental Insights from Comparative Approaches, B. Carlson, J. Sisneros, A. Popper, and R. Fay, Eds., New York: Springer Science+Business Media, LLC, 2019, vol. 70.D. Biswas, L. A. Arend, S. Stamper, B. P. Vágvölgyi, E. S. Fortune, and N. J. Cowan, "Closed-Loop Control of Active Sensing Movements Regulates Sensory Slip," Curr Biol, vol. 28, iss. 4, 2018.Perhaps the best connection of any to our work is the increase in active sensing with the loss of a sensory system – in this case perhaps the loss of a single eye would be analogous to the loss of a modality that causes increases to active sensing in a different modality. The change in active sensing strategy here is similar but suggests that the partial loss of a modality can lead to increases in active sensing in the same modality. Again, this connection is not critically important, just an observation that the authors may or may not find relevant to their discussion.

We appreciate the Reviewer’s suggestions and have incorporated more active sensing literature to the Discussion.

Reviewer #2 (Recommendations for the authors):1. The animal's success rate drops with gap distance. Is that caused by errors in vision (distance estimation) or jumping (motor accuracy). I wonder if the authors have data to address this question.

We have now incorporated additional experiments with larger landing platforms, which showed that failures at longer distances decreased relative to narrower platforms, suggesting that at longer distances the mice had difficulty landing on the relatively narrow platforms in the original experiments. In the new dataset, monocular animals showed no effect of gap distance on success rate, and binocular animals showed only a slight decrease in success with longer distances (p=0.48). These results have been incorporated into Figure 2.

2. It is true that mice can use monocular cues to estimate distance in this task, but the observed differences between monocular and binocular conditions (more head movements and the difference in the modeling results) actually suggest that mice normally use binocular cues. The authors should state this conclusion more prominently.

We agree, and think that mice will use all of the cues available, which under baseline conditions includes stereopsis. We did not mean to imply that they would not use binocular cues, but instead were focused on their ability to perform the task without binocular cues, and found this interesting given that monocular cues for depth are less well studied. We have updated the Discussion to include this point.

3. Related to the point #2 above, it would be interesting to know if the performance requires visual cortex under binocular condition.

We performed additional experiments with optogenetic suppression (binocular zone V1 suppression in binocular animals) and found a decrease in success rate, as well as changes in both pre-jump behavior (head angle) and landing position, consistent with V1 being utilized to perform the task under binocular conditions. Interestingly, animals with binocular vision showed no effect of binocular zone V1 suppression. These data have been incorporated in Figure 5.

4. The performance was impaired upon suppressing the visual cortex, but the animals can still do it. The authors discussed this as incomplete suppression. This is reasonable, but are there data to support this, like recording across the cortical depth and visual cortex? Related, the suppression was considered "V1", but how did the authors confirm that this was restricted to V1, and how much of V1 was illuminated by the light?

While we do not have electrophysiological recordings to characterize the spread of suppression, this has been done previously, and in cortex the PV-ChR2-mediated suppression is about 1 mm^3^ (Li et al., *eLife* 2019). Given the large volume of V1 (~5 mm^3^) it is unlikely we are completely suppressing all of V1. The changes in pre-jump behavior and landing position we have further characterized are consistent with the animals being unable to use the part of the visual field they normally would, and instead turning or lifting their heads to sample a different region of visual space that has not been affected by the manipulation. Furthermore, while binocular animals with binocular zone V1 suppression showed significant changes in behavior, monocular animals with binocular zone V1 suppression showed little change in behavior, consistent with the idea that they have already switched to using a different part of visual space after monocular occlusion.

Additional analyses have been included in Figure 5, and are further addressed in the Discussion.

Reviewer #3 (Recommendations for the authors):Since the development of a new task is a major part of the manuscript, I recommend more details in the Materials and methods for describing the task. Based on the current methods section, I assume the platform and distance were manually changed by the experimenter for each trial. Please provide more detail on the time it took for this manual step and discuss how this might or might not affect the animals' engagement in the task. Were the animals conditioned to this experimenter intervention? In addition, this labor-intensive step limits the number of trials mice can perform (30-60 per session). There should be information regarding the number of sessions each mouse performed. Related, I would like to see the behavior and optogenetics data from each animal plotted separately.

While manually placing the platforms takes some additional time, the bulk of the ‘down-time’ is between trials when mice can roam freely about the arena. Automating the system (which would be ideal) would allow for parallelization of experiments and more trials per session/longer sessions. We have added these additional details and discussion to the Methods section. We have also added the mean ± standard error for number of sessions and trials per session for each experiment to the Methods, and have plotted individual animal data on figure panels when practical.

If the claim of mice accurately estimated the distance was supported by the bottom panel of B and C on Figure 2, then please show both the jump distance for both the success trials and failed trials. Do mice often fall short on the failed trials compared to the success trials?What is the image inside of each circle in figure 3A? Is it a mouse? I am unclear how it is related to each movement state of the hidden Markov model. Please consider more informative illustrations. Related, I find the video showing the Deeplabcut markers used very helpful, please consider having a panel in figure 3 showing the markers feed into the hidden Markov model. This would also help make the example eye position traces in panel A clearer.The font of the numbers inside the matrix on figure 3B is too small, please make them bigger or remove them.

In order to measure accuracy, we performed additional experiments with larger landing platforms as described above. Under these conditions, mice only fail by jumping short. Having a larger potential landing area allows us to measure the accuracy and precision of the mice under the various experimental manipulations. We have also updated our decision period figure to show more intuitive example movement trajectories, and updated other panels per the Reviewer’s suggestions.

Most importantly, the conclusion from Figure 3 is Mice perform more head movements under monocular conditions. This conclusion wasn't intuitive from more transitions between states in the monocular condition. It is clear from the example traces that the states represent different movements, but more transition in states being the equivalent of more head movement would at least require example traces of head position around the transition in states.

We now use a more intuitive approach to movement identification and clustering than the ARHMM, and analyze the frequency (Figure 3D) and amplitude (Figure 3E) of these movements.

How many animals were used in the eye tracking experiments? The jump distance in Figure 4 panel C seems to be lower than the jump distance in Figure 2B and C, especially for the lower gap distance. Is this the strategy of the animals used in eye tracking, or is the eye tracking equipment weighing down the animals? Showing the success rate and jump distance of individual animals would be helpful.

The same animals were used for the eye tracking experiments as for the other experiments in the original version of the manuscript, thus we were able to directly compare their performance with the head-mounted hardware. There was not a significant change in performance. We have added these results and the number of mice to the Results section and updated the plots to include individual data.

As mentioned in the text, the change in performance during optogenetic suppression trials was largely due to an increase in abort with little change in failure rate. Therefore, showing an example video of an animal that failed a trial might be misleading.

We have found more exemplary videos to show the changes in precision with optogenetic V1 suppression.

Reference in the middle of the paragraph at Discussion, Eye movements during distance estimation section. Holmgren et al., is missing a date.

This preprint has since been published in a peer-reviewed journal, and we have updated the reference accordingly.